# Biopsychosocial approach to understanding determinants of depression among men who have sex with men living with HIV: A systematic review

Zul Aizat Mohamad Fisal[1]*, Halimatus Sakdiah Minhat[2], Nor Afiah Mohd Zulkefli[2], Norliza Ahmad[2]

1 Faculty of Medicine and Health Sciences, Universiti Putra Malaysia, Serdang, Selangor, Malaysia,
2 Department of Community Health Sciences, Faculty of Medicine and Health Sciences, Universiti Putra Malaysia, Serdang, Selangor, Malaysia

* zulaizat82@gmail.com

**Data Availability Statement:** All relevant data are within the paper.

## Abstract

### Introduction

Men who have sex with men (MSM) living with HIV are more likely to be depressed than MSM without HIV. The AIDS epidemic will not end if the needs of people living with HIV and the determinants of health are not being addressed. Compared to HIV individuals without depression, depressed HIV individuals have worse clinical outcomes and higher mortality risk. Depression is caused by a complex combination of social, psychological, and biological variables. This systematic review, thereby motivated by the need to address this gap in the literature, aims to articulate determinants of depression among MSM living with HIV according to the biopsychosocial approach.

### Methodology

We systematically searched four databases from 2011 to 2021. We searched for observational studies on determinants of depression among MSM living with HIV. The outcome is depression based on the categorical or numerical outcome. Two reviewers independently extracted data and assessed study risks of bias. Any disagreements are consulted with the third reviewer.

### Results

We identified 533 articles, of which only eight studies are included. A total of 3,172 MSMs are included in the studies. We found the determinants of depression and categorized them according to biological, psychological, and social approaches.

### Conclusion

The determinants of depression with the strongest evidence across studies were enacted HIV-related stigma, unemployment, sleep disturbance, current smoker, black ethnicity, born overseas, ART initiation, and access to mental health care. Despite weaker evidence, the

**Funding:** We received funding from our institution, which is Universiti Putra Malaysia (UPM). The funders had no role in study design, data collection, and analysis, decision to publish, or manuscript preparation. Authors received no salary from the funders.

**Competing interests:** NO

other relevant determinants to be included were older age, internalized stigma, self-efficacy, and social support. Efforts to improve or prevent depression among MSM living with HIV could benefit from addressing the determinants of depression based on the biopsychosocial approach immediately after HIV diagnosis. Integrating mental health screening and care into HIV treatment settings would strengthen HIV prevention and care outcomes and improve access to mental healthcare.

## 1. Introduction

Depression is marked by persistent sadness and a lack of interest or enjoyment in previously satisfying or pleasurable behaviors. It may also cause sleep and appetite disturbances, exhaustion, and poor concentration [1]. Based on the International Statistical Classification of Diseases and Related Health Problems, Tenth Revision (ICD–10), depression is diagnosed if the patient has two of the first three symptoms: 1) depressed mood; 2) loss of interest in everyday activities; and 3) reduction in energy, plus at least two of the remaining seven depressive symptoms [2]. An estimated 264 million people worldwide suffer from depression [1]. MSM living with HIV were more likely to be depressed compared to MSM without HIV. The previous extensive systematic reviews show that 40% to 43% of MSM living with HIV had depression [3, 4].

Compared to HIV patients without depression, depressed HIV patients have a worsened immune function, decreased adherence to antiretroviral therapy (ART), slower viral suppression, faster progression to AIDS, and a higher risk of mortality [5]. MSM are vulnerable to mental health problems due to sexual minority stress, society's homophobia, sexual orientation-based victimization, and stigmatization that hinder healthy behavior [6, 7]. Depression is caused by a variety of complicated interactions or multilevel mechanisms, including the following: 1) social factors such as socioeconomic status and social support; 2) psychological factors such as health beliefs and lifestyle and 3) biological factors such as physiological or genetic predispositions, and hence making the biopsychosocial (BPS) approach valid to investigate depression [1, 8, 9]. Concerning social factors, depression among MSM living with HIV was associated with physical abuse, being HIV positive, HIV risk behaviors [10], poverty and food insecurity [11, 12], and being unemployed [13]. The psychological factors of depression include verbal abuse [10], perceived stigma [14], low social support [15], and substance use [16]. Finally, the biological factors include younger age [17] and opportunistic infections [12]. The BPS model consists of the concept of psychobiological vulnerability, which is determined by risk factors such as biogenetic, psychological, somatic, and societal nature [18]. Therefore, the biopsychosocial approach will provide a holistic approach to finding the determinants of depression among MSM living with HIV.

Depression experienced by MSM living with HIV has physical, educational, social, financial, psychological, and short- and long-term health consequences [11]. The co-occurrence of HIV and depression was associated with poor health outcomes like poor quality of life and worsening disease states [19]. Untreated depression in MSM living with HIV can lead to risky sexual behavior, alcohol and drug misuse and abuse, and suicide [20]. Among MSM with newly diagnosed HIV, depression leads to poor adherence to antiretroviral drugs [21], resulting in poor immunological and virological outcomes [22].

According to the Sustainable Development Goals (SDGs) of the AIDS response, no one should be left behind. The AIDS epidemic will not end if the needs of people living with HIV (PLHIV) and the determinants of health are not being addressed [23]. The SDGs that firmly

address mental health include the following: 1) SDG 3: Good Health and Well-being; 2) SDG 5: Gender Equality; 3) SDG 10: Reduced Inequality; and 4) SDG 16: Peace, Justice, and Strong Institutions. The indicators are important because they encourage equity, justice, patient-centered care, community involvement, and mental health awareness [24]. The future of the HIV response is inextricably tied to worldwide efforts to combat non-communicable diseases (NCDs), including mental illness. The worldwide HIV response has increased the chances for gay and bisexual men groups to be recognized as citizens, holders of rights, and beneficiaries of public health programs in their nations, which is a distinct success [25].

Despite the growing HIV epidemic and depression among MSM worldwide and increased research in this population, empirical evidence on depression based on the biopsychosocial approach has not been synthesized. This systematic review, thereby motivated by the need to address this gap in the literature, aims to articulate determinants of depression among MSM living with HIV according to the biopsychosocial approach. It will offer recommendations for future mental and behavioral health interventions to reduce potential adverse outcomes in the HIV care continuum. Identifying determinants of depression will help the health authorities, stakeholders, and policymakers to provide a complete health package for MSM, in general, to support Ending AIDS 2030.

## 2. Methodology

This systematic review protocol was written following the Preferred Reporting Items for Systematic Review and PRISMA guidelines.

### 2.1 Eligibility criteria

Articles were included if they fulfill the following: 1) Published in English from 2011 to 2021; 2) used cross-sectional, cohort study, case-control study design; 3) used a standard instrument to assess depression; 4) Population: MSM living with HIV aged 18 and above; 5) Exposure: Factors associated with depression; 6) Comparison: MSM living with HIV without depression; and 7) Outcome: Depression. The review has depression determined by standard or accepted tools or instruments with categorical or numerical outcomes.

In contrast, articles were excluded if they were the following: 1) Review papers, conference abstracts, case reports, and qualitative studies, study protocol, mixed-methods study; 2) No statistical analysis conducted; and 3) If there is more than one study involving the same population, only the most recent published or comprehensive one will be included.

### 2.2 Information sources

Two reviewers independently searched four databases: 1) Academic Search Complete; 2) CINAHL; 3) Medline; and 4) SCOPUS from 3 May 2021 to 17 May 2021.

### 2.3 Search strategy

The following search terms were used: depression or depressive disorder or major depressive disorder AND HIV or Human Immunodeficiency Virus AND men who have sex with men or gay or homosexual or bisexual or MSM AND factors or determinants or predictors.

### 2.4 Study selection

Two reviewers independently screened the papers in two stages: title/abstract screening and full-text screening. The eligibility criteria were then compiled into a checklist. Then, we checked the titles and abstracts against the eligibility criteria. Next, we obtained all possibly

eligible articles' entire texts. The whole text was reviewed by two reviewers who applied inclusion criteria independently. When required, we resolved differences by consensus at both screening stages with the help of a third reviewer. All differences were recorded in Excel spreadsheets, along with the reasons for inclusion or exclusion.

## 2.5 Data collection process

We developed a data extraction sheet to guide data collection. This sheet directed us to collect the definition and methods for each step of the cascade, the results of estimations, and data sources. Two reviewers independently read each article and extracted the relevant data. In discussions with a third reviewer, any discrepancies in the extracted data were resolved by consensus.

## 2.6 Data item

The following data were extracted into two tables. The first table is the characteristic of selected articles: The first author/year, timing of data collection, study aim, study design, study location, and sample size. The second table is the determinants of depression according to the BPS approach: Author/year, screening tool, outcome definition (screening instrument cut-off or diagnostic criteria), and significant variables associated with depression and statistical value. When there were multiple estimates over time in the same study sample, the last one was chosen.

## 2.7 Quality assessment

The selected articles were entered into the quality assessment stage. Two authors independently assessed the quality of studies using the Joanna Briggs Institute (JBI) Appraisal Tools. There are two separate tools addressed specifically for cross-sectional and cohort studies. There are eight questions for the cross-sectional checklist, and 12 questions for the cohort checklist. Each question requires answering yes, no, unsure, or not applicable. Then, the decision was made on overall appraisal whether to include, exclude the studies, or seek further information [26]. The reviewers then met to discuss the results of their critical appraisal for the final appraisal. If the two reviewers disagree on the final critical appraisal and this cannot be resolved through discussion, the third reviewer was consulted. The authors then determined whether a study can be included, excluded, or seek further information.

## 2.8 Summary measures

The principal summary measures include multivariate analysis. The adjusted odds ratio (AOR) and beta values were taken with a 95% confidence interval. The significant value was chosen at $P<0.05$.

# 3. Results

## 3.1 Study selection

We identified 533 articles through our electronic databases and manual search, reducing them to 367 after removing duplicates. Then, 297 articles were excluded following titles and abstracts screening due to different study designs, not MSM living with HIV populations, and different outcomes. Then, 70 full-text articles were screened, of which 62 were excluded for the following reasons: different outcomes, the inclusion of HIV-negative participants in the study sample, and not MSM population in the study sample. Finally, eight studies were included in this review. Fig 1 shows the PRISMA flowchart.

**Fig 1. Preferred Reporting Items for Systematic Reviews and Meta-Analyses (PRISMA) flowchart for selecting studies.**

## 3.2 Studies characteristics

In the eight studies, there were a total of 3,172 MSM living with HIV. Four studies were conducted in China [27–30], two studies in the USA [31, 32], and one in Australia [33]. One study was conducted in two countries, which were the United Kingdom and Ireland [34]. The papers were published between 2015 and 2020. Six studies used a cross-sectional design, and two studies used a cohort design. The results are presented in Table 1.

## 3.3 Study quality

The appraisal results for the included studies are outlined in Tables 2 and 3. All studies included in the review got over 50% "yes" answers in the critical appraisal checklist. All outcomes in the studies included were measured in a valid and reliable way. One study had 75% "yes" [14], and one study had 83% "yes" [32] and the remaining studies had 100% yes based on the JBI checklist. Tables 2 and 3 show the quality assessment of cross-sectional and cohort studies.

## 3.4 Results of individual studies

Our review identified three domains for determinants of depression among MSM living with HIV: biological, psychological, and social. The results are presented in Table 4.

**Table 1. Characteristics of selected articles.**

| No | Author/year | Timing of data collection | Study aim | Study design | Study location | Sample size |
|---|---|---|---|---|---|---|
| 1. | Li et al. (2016) [27] | Not mentioned | To investigate the prevalence of depression and anxiety, and the significance of two risk factors (enacted HIV-related stigma and perceived stress) and one protective factor (gratitude) of depression/anxiety. | Cross-sectional | Chengdu, China | 321 |
| 2. | Tao et al. (2017) [28] | Not mentioned | To assess the relationship between HIV-related stigma and depression. | Cross-sectional | Beijing, China | 367 |
| 3. | Wang et al. (2019) [29] | March 2013 to March 2014. | To evaluate the relationship between self-efficacy and depression and anxiety. | Cross-sectional | Beijing, China | 367 |
| 4. | Luo et al. (2020) [30] | March 2013 to August 2014. | To determine the changes in mental health (depression and anxiety) one year after HIV diagnosis and the disparities in mental health trajectories. | Cohort | Changsha, China | 258 |
| 5. | Rood et al. (2015) [31] | Not mentioned | To investigate how different coping combinations may predict depression severity and the utilization of a range of clinically meaningful support services. | Cross-sectional | Massachusetts (USA) | 170 |
| 6. | Irwin et al. (2018) [32] | October 2001 to October 2012. | To determine an association between sleep disturbance and depression. | Cohort | Four sites in the USA: Baltimore, Maryland; Chicago, Illinois; Los Angeles, California; Pittsburgh, Pennsylvania. | 1054 |
| 7. | Heywood & Lyon (2016) [33] | August 2014 to December 2014. | To identify and compare risk and protective factors for depression, anxiety, and generalized stress. | Cross-sectional | Online recruitment in Australia | 357 |
| 8. | Murphy et al. (2018) [34] | May and November 2014 | To investigate the associations between forms of HIV-related optimism, HIV-related stigma, and anxiety and depression. | Cross-sectional | UK and Ireland | 278 |

**3.4.1 Biological.** Two studies found a significant association between depression and biological factors [30, 32]. Within the first year of diagnosis, ART initiation was associated with reduced depressive symptoms (β = -2.14) compared to those not placed on ART [30]. The other predictor with increased odds of depression was high viral load >10,000 copies/ml (OR: 1.38); however, the evidence is not strong [32]. Meanwhile, older age was associated with reduced odds of depression (OR: 0.98) [32].

**Table 2. Quality assessment of cross-sectional studies.**

| Questions | Li et al. (2016) [27] | Tao et al. (2017) [28] | Wang et al. (2019) [29] | Rood et al. (2015) [31] | Heywood & Lyon (2016) [33] | Murphy et al. (2018) [34] |
|---|---|---|---|---|---|---|
| 1. Were the criteria for inclusion in the sample clearly defined? | Y | Y | N | Y | Y | Y |
| 2. Were the study subjects and the setting described in detail? | Y | Y | N | Y | Y | Y |
| 3. Was the exposure measured in a valid and reliable way? | Y | Y | Y | Y | Y | Y |
| 4. Were objective, standard criteria used for measurement of the condition? | Y | Y | Y | Y | Y | Y |
| 5. Were confounding factors identified? | Y | Y | Y | Y | Y | Y |
| 6. Were strategies to deal with confounding factors stated? | Y | Y | Y | Y | Y | Y |
| 7. Were the outcomes measured in a valid and reliable way? | Y | Y | Y | Y | Y | Y |
| 8. Was appropriate statistical analysis used? | Y | Y | Y | Y | Y | Y |
| % Yes | 100 | 100 | 75 | 100 | 100 | 100 |

Abbreviations: Y = Yes; N = No; U = Unclear; NA = Not applicable.

**Table 3. Quality assessment of cohort studies.**

| Questions | Luo et al. (2020) [30] | Irwin et al. (2018) [32] |
|---|---|---|
| 1. Were the two groups similar and recruited from the same population? | Y | Y |
| 2. Were the exposures measured similarly to assign people to both exposed and unexposed groups? | Y | Y |
| 3. Was the exposure measured in a valid and reliable way? | Y | Y |
| 4.Were confounding factors identified? | Y | Y |
| 5. Were strategies to deal with confounding factors stated? | Y | Y |
| 6. Were the groups/participants free of the outcome at the start of the study (or at the moment of exposure)? | Y | Y |
| 7. Were the outcomes measured in a valid and reliable way? | Y | Y |
| 8. Was the follow up time reported and sufficient to be long enough for outcomes to occur? | Y | Y |
| 9. Was follow up complete, and if not, were the reasons to loss to follow up described and explored? | Y | N |
| 10. Were strategies to address incomplete follow up utilized? | Y | U |
| 11. Was appropriate statistical analysis used? | Y | Y |
| % Yes | **100** | **83** |

Abbreviations: Y = Yes; N = No.

**3.4.2 Psychological.**   Four studies reported a significant association between increased depression and stigma, either internalized HIV-related stigma [28, 33] or enacted HIV-related stigma [27, 34]. In our review, only enacted HIV-related stigma [27] is strongly associated with increased odds of depression (OR: 7.72), while the other three studies show weak evidence [13, 28, 32]. The other risk factors with increased odds of depression include sleep disturbance (OR: 1.52) and current smoker (OR:1.61) [32]. Meanwhile, weak evidence for increased depression has been found for engagement in high functional/high dysfunctional coping strategies and low functional/high dysfunctional coping strategies [31], perceived stress [27], and social stress [30]. With solid evidence, access to mental healthcare was associated with reduced depression ($\beta$ = -3.51) [30]. The other factors associated with reduced depression were gratitude [27], self-efficacy [29]) and HIV health optimism [34].

**3.4.3 Social.**   One study conducted in Australia found that unemployment was positively associated with increased depression ($\beta$ = 5.41) [33]. In another study, being born overseas was associated with reduced depression with good evidence ($\beta$ = -2.62), whereby black ethnicity was associated with increased odds of depression (OR: 1.62) [32]. Increases in social support were associated with decreased depressive symptoms, but the evidence is weak [30].

Regarding outcome measurement of depression, three studies [27, 31, 32] used the Center for Epidemiological Studies-Depression checklist (CES-D). Following the test objectives, the CES-D provides cut-off scores; for example, a score of 16 or higher can aid in identifying persons who are at risk for clinical depression, with good sensitivity and specificity, as well as a high level of internal consistency [35, 36]. Another three studies used the Hospital Anxiety and Depression Scale (HADS) [28, 29, 34]. HADS focuses on non-physical symptoms, so that it can be used to diagnose depression [37]. The remaining study used the Patient Health Questionnaire-9 (PHQ-9) [30] and the Depression Anxiety Stress Scale-21 (DASS-21) [33]. The PHQ-9 can be used for diagnostics as well as a depression severity score [38], while the DASS-21 satisfactorily predicts depression as diagnosed with the Mini International Neuropsychiatric Interview (MINI) [39].

**Table 4.** Determinants of depression according to the biopsychosocial approach.

| No | Author/year | Screening tool | Outcome definition of depression | Significant variables associated with depression | | Statistical value |
|---|---|---|---|---|---|---|
| 1. | Li et al. (2016) [27] | 20-item Center for Epidemiological Studies-Depression (CES-D) scale | 16/21/25 for mild, moderate, and severe depression | **Psychological** | Perceived stress | AOR: 1.17, 95% CI = 1.12, 1.22, P = 0.001 |
| | | | | | Enacted stigma | AOR: 7.72, 95% CI = 2.27, 26.25, P<0.001 |
| | | | | | Gratitude | AOR: 0.90, 95% CI = 0.86, 0.94, P<0.001 |
| 2. | Tao et al. (2017) [28] | Hospital Anxiety and Depression Scale (HADS) | A score of 0 to 7 was defined as normal, 8 to 10 as borderline depression, and a score of 11 to 21 as suspected depression. | **Psychological** | Internalized stigma | AOR: 1.09, 95% CI: 1.07, 1.12, P<0.001. |
| | | | | | Vicarious stigma from the community/health care | AOR: 1.06, 95% CI: 1.03, 1.10, P<0.001 |
| 3. | Wang et al. (2019) [29] | Hospital Anxiety and Depression Scale (HADS) | A score of 0 to 7 was defined as normal, 8 to 10 as borderline depression, and a score of 11 to 21 as suspected depression. | **Psychological** | Self-efficacy | AOR: 0.88, 95% CI: 0.85, 0.92, P<0.001 |
| 4. | Luo et al. (2020) [30] | Patient Health Questionnaires Depression Scale (PHQ-9) | A score of 10 the cut-off score for significant depressive symptoms | **Biological** | Received ART during the first year after diagnosis. | β = −2.14, P = 0.008 |
| | | | | **Psychological** | Participants who had access to mental health care after diagnosis were more likely to improve depression. | β = −3.51, P = 0.003 |
| | | | | | Increases in social stress scores were associated with increases in depression. | β = 0.43, P<0.001 |
| | | | | **Social** | Increases in support were associated with decreases in PHQ-9 score. | β = −0.37, P<0.001 |
| 5. | Rood et al. (2015) [31] | Center for Epidemiological Studies-Depression (CES-D) scale | A total score ranging from 0 to 60, and a clinical cut-off score of 23, instead of 16, was used to indicate probable depression. | **Psychological** | High Functional/High Dysfunctional coping strategies | β = 0.36, t = 4.47, P< 0.01 |
| | | | | | Low Functional/High Dysfunctional coping strategies | β = 0.50, t = 6.34, P< 0.01 |
| 6. | Irwin et al. (2018) [32] | Center for Epidemiological Studies-Depression (CES-D) scale | A score≥ 16 represents a higher risk of depression. | **Biological** | Older age | OR: 0·98, 95% CI: 0·96, 0·99, P<0.05 |
| | | | | | Viral load > 10,000 copies/ml | OR: 1·38, 95%CI: 1·04, 1·85, P<0.05 |
| | | | | **Psychological** | Sleep disturbance | OR: 1·52, 95%CI: 1·29, 1·80, P<0.001 |
| | | | | | Current smoker | OR: 1·61, 95% CI: 1·12, 2·33, P<0.05 |
| | | | | **Social** | Black ethnicity | OR: 1·62, 95% CI: 1·17, 2·24, P<0.05 |
| 7. | Heywood & Lyon. (2016) [33] | The short-form Depression Anxiety Stress Scales (DASS-21) | A higher score represents a greater indication of depression. | **Psychological** | Experiencing greater internalized stigma | β = 1.14, P<0.001 |
| | | | | **Social** | Unemployment | β = 5.41, P = 0.05 |
| | | | | | Born overseas | β = − 2.62, P = 0.05 |

(*Continued*)

**Table 4.** (Continued)

| No | Author/year | Screening tool | Outcome definition of depression | Significant variables associated with depression | | Statistical value |
|----|-------------|----------------|----------------------------------|-------------------------------------------------|--|-------------------|
| 8. | Murphy et al. (2018) [34] | 14-item Hospital Anxiety and Depression Scale (HADS) | A score of 0 to 7 was defined as normal, 8 to 10 as borderline depression, and a score of 11 to 21 as suspected depression. | **Psychological** | HIV Health Optimism | β = − 0.15, 95% CI: -0.44, -0.06, P<0.05 |
| | | | | | Enacted stigma | β = 0.15, 95% CI: 0.02, 0.28, P<0.05 |
| | | | | | Internalized stigma | β = 0.36, 95% CI: 0.26, 0.09, P<0.001 |

Abbreviations: AOR: Adjusted odds ratio; OR: odds ratio, CI: confidence interval.

## 4. Discussion

Our review found that the potential determinants in depression among MSM living with HIV varied across studies. However, the determinants of depression with good evidence are enacted HIV-related stigma, unemployment, sleep disturbance, current smoker, black ethnicity, born overseas, ART initiation, and access to mental health care. The other relevant determinants to be discussed are older age, internalized stigma, self-efficacy, and social support. Although the evidence may be weak or moderate for some, they are relevant to be considered and analyzed for theoretical or practical reasons.

### 4.1 Biological factors

**4.1.1 Age.** One study shows older age was a protective factor for depression (OR:0.98) [32]. The age factor should be highlighted due to the increased prevalence of young MSM living with HIV [40] and the association of depression among MSM with young age [41]. Depression was prevalent among adolescents and young adults with illnesses such as HIV, underscoring the importance of improved psychological examination and monitoring, particularly on young patients [42].

**4.1.2 Antiretroviral therapy (ART).** Initiation of ART in the first year after HIV diagnosis was associated with reducing depressive symptoms in one study [30]. The finding is in line with other studies, whereby PLHIV placed on ART had reduced depression and higher quality of life (QOL) for physical, psychological, and environmental domains [43–45]. Accelerated commencement of ART has been shown to improve clinical results, and follows the WHO recommendation in supporting accelerated ART initiation, including same-day ART initiation [46]. During acute HIV infection, ART limits the viral reservoir, preserves immune function, and decreases systemic inflammation [47]. The introduction of ART has reduced morbidity, mortality and increased the quality of PLHIV [48]. Furthermore, Treatment as prevention (TasP) which uses ART among HIV-positive persons to decrease the chances of HIV transmission will benefit all serodiscordant couples [49]. With all the shreds of evidence of ART benefits, early initiation of ART could give MSM reassurance towards achieving a good quality of life that could reduce their risk of having depression or improve the state of depression.

### 4.2 Psychological factors

**4.2.1 Enacted HIV-related stigma.** Enacted HIV-related stigma refers to the experience of being discriminated, stigmatized, and treated in an unfriendly manner due to their HIV status, and it is often influenced by the perception of others' attitudes towards PLHIV [27, 28]. Sixty-nine countries, nearly half of which are in Africa, have laws criminalizing homosexuality

[50]. In addition, MSM could see themselves as doubly stigmatized due to their sexual identity and HIV status [51, 52]. Three studies in this review [27, 28, 34] show enacted HIV-related stigma as the determinant of depression among HIV MSM. The strongest evidence came from an adjusted odds ratio of 7.72, in which participants judged the degree of enacted HIV-related stigma based on personal experiences of discrimination, stigmatization, or unfriendly treatment because of their HIV diagnosis [27]. Previous systematic reviews and studies were consistent with the enacted HIV-related stigma associated with poor mental health and depression among PLHIV [53–55].

Previous studies show that MSM living with HIV face discriminatory treatment from the general public, MSM community, and the healthcare system [56–58]. Stigma and social pressure for MSM also come from their families who urge them to marry and have children to protect their family reputation and lineage [59]. In general, minority stress explains that stigma, prejudice, and discrimination create a hostile and stressful social environment that causes mental health problems [60]. Therefore, health care providers need to address stigma in their facilities and educate patients' family members or their circle of confidentiality.

**4.2.2 Internalized HIV-related stigma.** Internalized HIV-related stigma refers to endorsing negative beliefs and feelings about oneself because of one's HIV-positive status [28]. Although evidence is not strong, three studies mentioned it as a risk factor for increased depression [28, 33, 34]. Internalized HIV-related stigma was positively associated with depression, which, combined with enacted HIV-related stigma, increased the likelihood of suicide [61]. In another study, internalized HIV-related stigma partly mediated the relationship between enacted HIV-related stigma and depression symptoms [62]. When combined with enacted HIV-related stigma by health care practitioners, stigma can impede MSM utilization of health services; thus, stigma should be addressed at individual and institutional levels [63].

**4.2.3 Self-efficacy.** Self-efficacy is the belief in one's ability to plan and carry out the actions necessary to manage potential scenarios [64]. Self-efficacy is a protective factor for depression in one study (OR:0.88) [29]. Lack of coping self-efficacy among HIV individuals may increase the likelihood of non-disclosure and depression [65]. In addition, self-efficacy for HIV disclosure decisions is the novel stressor for newly diagnosed individuals [66]. Therefore, interventions that enhance self-efficacy may help manage the demands of daily life with HIV, increase disclosure, and reduce depression [63].

**4.2.4 Sleep disturbance.** Sleep disturbance is associated with increased depression among MSM living with HIV [32]. The finding was consistent with the previous study among PLHIV, whereby those with more sleep problems were significantly more likely to have worse depression over time than those with fewer sleep problems [67]. Poor sleep quality was observed in 47% to 73% of PLHIV [68, 69]. Sleep disturbances can occur early in the course of HIV infection and suggest that the observed changes in sleep physiology could result from central nervous system involvement or immune defense mobilization in the early phases of HIV infection [70]. In addition, factors that influence sleep include psychosocial factors such as stigma and social isolation [66]. Therefore, it is imperative to identify and treat sleep disturbances in MSM living with HIV to improve mental health and quality-of-life outcomes.

**4.2.5 Current smoker.** One study found an association between current smokers and depression among MSM living with HIV [32]. The finding was observed in another study where HIV-infected smokers reported higher symptoms of depression than non-smokers [71]. In Canada, a study among the lesbian, bisexual, gay, transgender, and questioning (LBGTQ) population who smoke regularly showed that 62% suffer from depression symptoms, and 38% had a major depressive disorder [72]. On the other hand, they are more likely to smoke and use drugs to cope with stress, illness, social disadvantages, and sexual orientation concerns, all of which can be barriers to quitting smoking [72]. Apart from non-communicable diseases,

PLHIV who smoke cigarettes are more likely to get an opportunistic infection like oral candidiasis, Pneumocystis pneumonia, and gastrointestinal infection [73, 74]. Smoking was linked to more than 60% of fatalities in PLHIV, whereby they lose more life-years due to smoking, with 12.3 years lost to smoking than 5.1 years lost to HIV [75]. All the physical impacts of smoking may indirectly contribute to depression among MSM. Therefore, health care providers must recognize smoking behavior among MSM and make smoking cessation a priority for them.

**4.2.6. Access to mental health care.** One study conducted in China shows that participants who had access to mental health services were more likely to experience reduced depression [30]. Among Chinese MSM, minority stress is a significant predictor of psychological distress. Nonetheless, mental health treatments and interventions concentrating on MSM are lacking in China [76]. Previous systematic reviews and meta-analyses demonstrate that PLHIV can benefit from various mental and behavioral health interventions [77, 78]. MSM are more likely than other men to have tried to commit suicide and have succeeded in suicide [79]. As MSM suffer double discrimination for being gay and living with HIV, it could prevent them from accessing mental health services. Therefore, early and routine screening is mandatory to provide them with early mental health support and treatment intervention.

## 4.3 Social factors

**4.3.1 Social support.** Social support is a significant factor for improvement in depression in one study in this review [30]. Social support has been linked to coping with life stressors such as HIV/AIDS and other chronic health issues [80]. Evidence suggests that improved social support may reduce depressive symptoms among MSM living with HIV [81]. Another study found that the negative association between social isolation and depression was stronger for sexual minority male youths than non-minority youths and sexual minority females [82]. MSM with more social support networks also perceived lower levels of stigma [27]. MSM are prone to inadequate social support and severe depression symptoms, emphasizing the importance of developing psychological interventions specifically for them [83].

**4.3.2 Unemployment.** Depression scores increased among MSM living with HIV in unemployed men [33], as supported by another study [84]. PLHIV may encounter challenges to employment and retention due to HIV-related stigma, disclosure and confidentiality difficulties at work, the impact of poor health on their capacity to work, and the requirement for medical leave and healthcare appointments [85]. Additionally, unemployed PLHIV are less likely to have a sufficient income, a meaningful life, a daily routine, social support, and a sufficient income and participation in intellectually engaging activities [86, 87]. Therefore, psychosocial treatment combined with early ART could serve MSM living with HIV mentally and physically fit in sustaining their job and avoiding unemployment.

**4.3.3 Born overseas.** Lower depression scores were found among migrants in one study conducted in Australia [33]. Many studies have observed a "healthy migrant effect" (HME) among PLHIV, for example, in North America, Denmark, Germany, Spain, and Norway [88]. Based on HME, migrants often have a better health status than the remaining population in the native country, compared with the majority in the host country, especially during the first five to ten years after immigration [88]. In contrast, stigma and lack of access to care appear to be the main factors influencing poor HIV outcomes among migrants in high-income countries [89]. With regards to mental health among migrants, the two widely accepted theories were the following: (1) Loneliness theory: to escape loneliness, migrants frequently reside with other migrants, particularly those from the same country of origin [90]; and (2) Acculturation theory: owing to their experiences with discrimination and rejection, migrants feel like they are

not part of the community [91]. Therefore, improving migrants' access to health and HIV care will require a human rights-based approach to the governance of the entire migration process [92].

**4.3.4 Black ethnicity.** Black ethnicity was the predictor of increased risk of depression in this review [32]. The finding is consistent with the previous studies that depression may affect this group more than the general adult population due to racism, homophobia, and other forms of discrimination; however, it has been less fully explored [93–96]. Likewise, racial disparities in ART initiation for black MSM have been reported in the UK and the USA [97]. Despite rising interest in understanding how social factors contribute to poor health outcomes, many academics and relevant bodies remain wary of naming racism a primary cause of racial health disparities [98]. In keeping with SDG 10, inequalities in health care of any kind based on race, ethnicity, origin, or religion should be addressed by diverse stakeholders' commitment.

## 5. Limitations

Inconsistency and limited comparability of results may have been caused by the heterogeneity of outcome assessments using different tools to measure depression and different statistical analyses. Furthermore, our review involved studies from different countries. As a result, patient demographic factors also substantially affected the outcome. It also implies different cultural and legal contexts, with profound variations in accepting gay and bisexual (GB) identities and rights. This influences the levels of social stigma, which is one of the main factors identified in this systematic review. More importantly, some of these countries have laws protecting GB people's rights, whereas others do not.

Most of the studies were cross-sectional. It has weaknesses like difficulty in making a causal inference. Associations identified might be challenging to interpret and cannot investigate the temporal relationship between outcomes and risk factors. We did not include non-English language articles, which may have introduced bias as most studies come from English-speaking countries, and there is little evidence from other regions.

## 6. Conclusion

The determinants of depression with the strongest evidence among MSM living with HIV were enacted HIV-related stigma, unemployment, sleep disturbance, current smoker, black ethnicity, born overseas, ART initiation, and access to mental health care. Although the evidence may be weak or moderate, other risk factors worth considering are older age, internalized HIV stigma, self-efficacy, and social support. Efforts to improve and prevent depression among MSM living with HIV could benefit from addressing these determinants based on the biopsychosocial approach as soon as possible after HIV diagnosis. Engagement and retention in care will improve medical outcomes, in line with Ending AIDS 2030 and the SDGs that aim to leave no one behind.

## 7. Recommendations

Integrating mental health screening and care into HIV treatment settings would strengthen HIV prevention and care outcomes and improve access to mental healthcare. Actions to fight HIV-related stigma, in general, should be prioritized, as it is one of the key determinants of depression among MSM. A structured program on eliminating HIV-related stigma should be planned at multiple levels, including the interpersonal, institutional, community, and judicial levels. Legislations protecting the rights of GB citizens should be enacted immediately; as of present, they are struggling for their civil rights in a forum, in courtrooms, and on the streets.

Healthcare services should be designed to acknowledge, affirm, and validate diverse sexual identities. A solid and sustainable support system could prevent depression or aid in managing depression. The other determinants of depression, such as age, unemployment, self-efficacy, sleep disturbance, and smoking behavior, need to be addressed when providing care for MSM. Elimination of disparities for black MSM and migrants needs to address structural barriers or differences in HIV clinical care access and outcomes. More importantly, ART should be provided as soon as possible, aligning with the SDGs to treat all irrespective of immune status. Additional research is needed to better understand depression and minority stress theory among MSM living with HIV and how interventions could be tailored to meet specific needs.

## Supporting information

**S1 Checklist.**
(DOCX)

## Author Contributions

**Conceptualization:** Zul Aizat Mohamad Fisal, Halimatus Sakdiah Minhat, Nor Afiah Mohd Zulkefli.

**Data curation:** Zul Aizat Mohamad Fisal, Halimatus Sakdiah Minhat.

**Formal analysis:** Zul Aizat Mohamad Fisal, Halimatus Sakdiah Minhat.

**Methodology:** Zul Aizat Mohamad Fisal, Halimatus Sakdiah Minhat, Nor Afiah Mohd Zulkefli, Norliza Ahmad.

**Supervision:** Nor Afiah Mohd Zulkefli, Norliza Ahmad.

**Writing – original draft:** Zul Aizat Mohamad Fisal.

**Writing – review & editing:** Zul Aizat Mohamad Fisal, Halimatus Sakdiah Minhat, Nor Afiah Mohd Zulkefli, Norliza Ahmad.

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
