## [Decision Letter · Decision Letter 0]

9 Nov 2021

PONE-D-21-24036Application of biopsychosocial construct to understand determinants of depression among men who have sex with men living with HIV: A systematic reviewPLOS ONE

Dear Dr. Mohamad Fisal,

Thank you for submitting your manuscript to PLOS ONE. After careful consideration, we feel that it has merit but does not fully meet PLOS ONE’s publication criteria as it currently stands. Therefore, we invite you to submit a revised version of the manuscript that addresses the points raised during the review process. Below you can find the comments raised by the reviewer. Please read it carefully, and edit your manuscript accordingly, with particular focus to the methodological comments. 

We look forward to receiving your revised manuscript.

Kind regards,

Omar Sued, MD

Academic Editor

PLOS ONE

“NO”

“NO”

5. Please include your tables as part of your main manuscript and remove the individual files.

7. We noticed you have some minor occurrence of overlapping text with the following previous publication(s), which needs to be addressed:

- https://journals.lww.com/aidsonline/Fulltext/2019/07150/Mental_health_and_HIV-AIDS__the_need_for_an.1.aspx

- https://www.wjpps.com/Wjpps_controller/abstract_id/12341

In your revision ensure you cite all your sources (including your own works), and quote or rephrase any duplicated text outside the methods section. Further consideration is dependent on these concerns being addressed.

Reviewers' comments:

Reviewer's Responses to Questions

**Comments to the Author**

1. Is the manuscript technically sound, and do the data support the conclusions?

Reviewer #1: Partly

2. Has the statistical analysis been performed appropriately and rigorously? 

Reviewer #1: N/A

3. Have the authors made all data underlying the findings in their manuscript fully available?

Reviewer #1: No

4. Is the manuscript presented in an intelligible fashion and written in standard English?

Reviewer #1: No

5. Review Comments to the Author

Reviewer #1: I appreciate the opportunity to review this manuscript that approaches a highly relevant topic for public health, such as correlates of depressive symptoms among MSM with HIV. This review is necessary as a systematization of evidence that can guide the development of public policies and interventions, to improve mental health and HIV outcomes in this population. I congratulate the authors for choosing this topic and for a remarkable work. However, I consider that there is still room for improvement of this manuscript so that it is suitable for publication.

General overview

-Several issues were found regarding writing, grammar and spelling throughout the whole manuscript (e.g., typos, syntactic and grammar errors, missing words, etc.). A thorough proofread is required.

-It is noteworthy that citation style changes throughout the manuscript. Adequacy to the journal requirements and guidelines should be thoroughly revised.

Abstract

-In the first sentence, I recommend using the present tense “are” rather than the past tense “were”.

-The first sentence in the Conclusions paragraph would be better located in the Results paragraph.

Introduction

-The use of the term “gay” is recommended instead of the term “homosexual”. In paragraph 5 the term “HIV-seropositive MSM” is used. I recommend using “MSM living with HIV” which is actually the term used throughout the manuscript, for consistency.

-Paragraph 3, sentence 2: It should be “MSM are vulnerable…”

-The introduction may benefit from a clearer order and organization of information. At some parts, it is somewhat repetitive and there is some disconnection between paragraphs. A possible reorganization is, for example, the following: current paragraph 1 is fine as an introduction of the main variable, second paragraph should be a summary of determinants of depression both in the general population and among MSM and people with HIV and presentation of the BPS approach (which is currently distributed between paragraphs 3 and 4, both paragraphs could be merged and integrated), third paragraph should be about consequences of depression among MSM with HIV (currently paragraph 5), the fourth paragraph could be a conclusion about the importance to address depression among MSM living with HIV (currently paragraph 2), the sixth paragraph is fine with conclusion and objectives.

-In the Introduction, the BPS construct is introduced as BPS approach, which I find it is a better term than construct. The authors may consider using the term BPS approach also in the rest of the manuscript.

Methods

-Quality assessment: The remaining articles are the selected articles?

Results:

-Study selection: reasons for exclusion, as listed in this section, should be expressed in a clearer way so that readers can accurately understand why a set of articles was excluded from analysis. For example, one reason is “the general PLHIV population”. That would not be the reason exactly, but “inclusion of general PLHIV population or not MSM population in the study sample”.

-Study characteristics: “Studies’ characteristics” would be a more appropriate title for this section as it describes the characteristics of the studies included in this review, and not the characteristics of the review itself.

-As previously mentioned, I recommend using the term “MSM living with HIV” or “MSM with HIV”, instead of HIV positive or seropositive MSM (or simply HIV MSM, as in the Discussion, please avoid using this term). I recommend consistency in the use of terms.

-I also recommend using the term “social” instead of “sociological”, as it is not related to sociology but to society.

-Within the biological factors, it is stated that “ART initiation improved depression”. This means that depression was reduced, it decreased, it was associated with a reduction of depressive symptoms? Perhaps the sentence could be expressed in a clearer way. The same for viral load, it is not clear if it is associated with increments or reduction of depressive symptoms. I suggest expressing the relations between factors and depression in a clearer way, indicating if they are associated with increased or reduced odds of depression. This same recommendation applies for psychological and social factors.

-Within the psychological factors, it is stated that stigma is associated with increased odds of depression. However, it is not mentioned what kind of stigma: HIV-related stigma, stigma related to sexual orientation (being gay or bisexual) or other kind? I recommend clarifying this.

-Regarding the CES-D, please revise the correct name of the instrument for greater accuracy.

-Some of the instruments listed measure “presence of depressive symptoms” and not “depression” (which is a specific disorder and diagnosis). That is, they screen for depressive symptoms but do not allow to arrive to a diagnosis of depression. This is the case, for example, of the CES-D. In that sense, it can be questioned that the present study is actually systematizing and reviewing factors associated with presence of depressive symptoms, and not factors associated with depression (which is not the same).

Discussion

-Among the biological factors, older age was a protective factor. Given the vulnerability that younger people exhibit and that it is also a key population regarding HIV and mental health, this is a result that is worth mentioning and explaining in the discussion section.

-In the “enacted stigma” subsection, it is not clear what “same-sex sexual identity” means, I understand that the authors meant simply “sexual identity”. In this section, it should be clarified what kind of stigma studies refer to.

-In the “current smoker” section, it is stated that “The number of life-years lost is…, respectively”. However, it is not clear “respectively” to what or who is referring.

-There are other social factors that are worth mentioning and discussing which have not been included in the Discussion section, such as unemployment and migration. In general, I recommend expanding the Discussion with the inclusion of other relevant factors that have been identified as determinants of depressive symptoms (e.g., unemployment, migration, age, self-efficacy, social support). Although evidence may be weak or moderate for some of them, they are relevant to be considered and analyzed for theoretical or practical reasons.

-Limitations: I would not say “incomparability”, but “limited comparability”, as studies were actually compared.

-The fact that studies from different countries were included not only implies demographic differences between patients. It also implies different cultural and legal contexts, with deep variations in the acceptance of gay and bisexual (GB) identities and rights. This influences the levels of social stigma (one of the main factors identified in this study). Some of these countries have laws protecting GB people’s rights whereas others do not. This should be acknowledged.

-“Foreign language” refers to non-English language? Foreign language depends on the native language of the authors and readers. I agree that this choice introduced bias as most studies come from English speaking countries and there is little evidence from other regions. I would explicitly mention this.

-Conclusion: Regarding the first sentence, the determinants that were identified were more than those that are mentioned. I would mention that these were the determinants with the strongest evidence.

-Recommendations: I suggest mentioning more explicitly, actions to fight stigma in general (e.g., enactment of laws to protect GB people’s rights) and in healthcare services (e.g., services that acknowledge, affirm, and validate diverse sexual identities). Elimination of stigma is mentioned too broadly in the recommendations perhaps, and it deserves a more detailed and expanded mention as one of the key determinants of depression among MSM.

-Table 1, study 4: the date is incomplete.

-Table 1, study 1: the aim is to study predictors of anxiety and generalized stress. What was the role of depression in this study and why was it included if depression was not the outcome?

-In table 2, I recommend placing the studies in the same order as Table 1 to facilitate integration of the information from both tables by the reader, as both tables are actually related.

-According to Table 2, many factors or determinants of depression that were identified in the present review were drawn from the study of Irving et al. (2018). For that reason, one would argue that the integration of results across studies was limited, and that this manuscript is, in some sense, a summary or reproduction of results of other few studies. Relevant factors that are mentioned in other studies (e.g., internalized stigma, which is mentioned in 3 studies) are barely taken into consideration and analyzed in this review (despite being a very important, repeatedly mentioned factor). A greater systematization of findings is required so that this manuscript is not a mere reproduction of results of a few studies, because that adds little value and significant contribution to the field.

6. PLOS authors have the option to publish the peer review history of their article (what does this mean?). If published, this will include your full peer review and any attached files.

Reviewer #1: No

---

## [Author Response · Author response to Decision Letter 0]

10 Jan 2022

Dec 15, 2021 

Dear PLOS editors,

Rebuttal Letter to manuscript:

Biopsychosocial approach to understanding determinants of depression among men who have sex with men living with HIV: A systematic review

Zul Aizat Mohamad Fisal1, Halimatus Sakdiah Minhat2, Nor Afiah Mohd Zulkefli2, Norliza Ahmad2

1DrPH Candidate, Faculty of Medicine and Health Sciences, Universiti Putra Malaysia, 43400 Serdang, Selangor, Malaysia

2Department of Community Health Sciences, Faculty of Medicine and Health Sciences, Universiti Putra Malaysia, 43400 Serdang, Selangor, Malaysia

Dear Academic Editor Omar Sued and dear reviewers,

We would like to thank you for your comments and recommendations, which gave us the opportunity to improve the paper. In the updated manuscript, we expect to answer all the issues identified. In this document we answer to all the questions highlighted by the reviewers.

Comments are shown in bold font, followed by our answer/comment in normal font. The corrections/changes in the manuscript are displayed through the track changes.

Editor’s comment:

Dear editor, we thank you for the general appreciation of our work and the specific comments given that help to improve our manuscript. The manuscript has followed all the journal requirements.

“NO”

Dear editor, there is no grant for our work, but we received funding from our institution, which is Universiti Putra Malaysia (UPM).

Dear editor, the funders had no role in study design, data collection, and analysis, decision to publish, or manuscript preparation.

Dear editor, authors received no salary from the funders.

Done. Thank you.

“NO”

Done. Thank you.

Dear editor, all data are fully available without restriction. Thank you.

5. Please include your tables as part of your main manuscript and remove the individual files.

Done. Thank you.

Dear editor, the reference list has been corrected and completed.Thank you.

7. We noticed you have some minor occurrence of overlapping text with the following previous publication(s), which needs to be addressed:

-https://journals.lww.com/aidsonline/Fulltext/2019/07150/Mental_health_and_HIV-AIDS__the_need_for_an.1.aspx

- https://www.wjpps.com/Wjpps_controller/abstract_id/12341

In your revision ensure you cite all your sources (including your own works), and quote or rephrase any duplicated text outside the methods section. Further consideration is dependent on these concerns being addressed.

Dear editor, all minor occurrence of overlapping text has been addressed. Thank you.

Reviewer’s comment:

1. Is the manuscript technically sound, and do the data support the conclusions?

Reviewer #1: Partly

Dear reviewer #1, we thank you for the general appreciation of our work, and specific comments given that help to improve our manuscript. The conclusions has been drawn appropriately based on the data presented.

2. Has the statistical analysis been performed appropriately and rigorously?

Reviewer #1: N/A

3. Have the authors made all data underlying the findings in their manuscript fully available?

Reviewer #1: No

Dear reviewer #1, all data are fully available without restriction. Thank you.

4. Is the manuscript presented in an intelligible fashion and written in standard English?

Reviewer #1: No

Dear reviewer #1, the typographical or grammatical errors has been corrected at revision. The revised manuscript has been proofread by certified proof readers.

5. Review Comments to the Author

Reviewer #1: I appreciate the opportunity to review this manuscript that approaches a highly relevant topic for public health, such as correlates of depressive symptoms among MSM with HIV. This review is necessary as a systematization of evidence that can guide the development of public policies and interventions, to improve mental health and HIV outcomes in this population. I congratulate the authors for choosing this topic and for a remarkable work. However, I consider that there is still room for improvement of this manuscript so that it is suitable for publication.

Dear reviewer #1. Thank you. We try to improve and do our best.

General overview

-Several issues were found regarding writing, grammar and spelling throughout the whole manuscript (e.g., typos, syntactic and grammar errors, missing words, etc.). A thorough proofread is required.

-It is noteworthy that citation style changes throughout the manuscript. Adequacy to the journal requirements and guidelines should be thoroughly revised.

Dear reviewer #1. The revised manuscript has been proofread by certified proof readers. The citation erros has been corrected based on Vancouver style.

Abstract

-In the first sentence, I recommend using the present tense “are” rather than the past tense “were”.

Dear reviewer #1, we have changed “were” to “are”.

-The first sentence in the Conclusions paragraph would be better located in the Results paragraph.

Dear reviewer #1, the sentence has been locted in result paragraph.

Introduction

-The use of the term “gay” is recommended instead of the term “homosexual”. In paragraph 5 the term “HIV-seropositive MSM” is used. I recommend using “MSM living with HIV” which is actually the term used throughout the manuscript, for consistency.

Dear reviewer #1, the term “MSM living with HIV” has been used throughout the manuscript, for consistency.

-Paragraph 3, sentence 2: It should be “MSM are vulnerable…”

Dear reviewer #1, we have followed you suggestion here.

-The introduction may benefit from a clearer order and organization of information. At some parts, it is somewhat repetitive and there is some disconnection between paragraphs. A possible reorganization is, for example, the following: current paragraph 1 is fine as an introduction of the main variable, second paragraph should be a summary of determinants of depression both in the general population and among MSM and people with HIV and presentation of the BPS approach (which is currently distributed between paragraphs 3 and 4, both paragraphs could be merged and integrated), third paragraph should be about consequences of depression among MSM with HIV (currently paragraph 5), the fourth paragraph could be a conclusion about the importance to address depression among MSM living with HIV (currently paragraph 2), the sixth paragraph is fine with conclusion and objectives.

Dear reviewer #1, we have followed you suggestion here. The second paragraph is the summary of determinants of depression both in the general population and among MSM and people with HIV and presentation of the BPS approach (Paragraphs 3 and 4, has been merged and integrated and become the second paragraph). The third paragraph are the consequences of depression among MSM with HIV. The fourth paragraph is the conclusion about the importance to address depression among MSM living with HIV (taken from paragraph 2). 

-In the Introduction, the BPS construct is introduced as BPS approach, which I find it is a better term than construct. The authors may consider using the term BPS approach also in the rest of the manuscript.

Dear reviewer #1, the term BPS approach has been used in the rest of the manuscript including in the title.

Methods

-Quality assessment: The remaining articles are the selected articles?

Dear reviewer #1, yes, the remaining articles are the selected articles. The sentence has been corrected.

Results:

-Study selection: reasons for exclusion, as listed in this section, should be expressed in a clearer way so that readers can accurately understand why a set of articles was excluded from analysis. For example, one reason is “the general PLHIV population”. That would not be the reason exactly, but “inclusion of general PLHIV population or not MSM population in the study sample”.

Dear reviewer #1, the reasons for exclusion has been changed to: the inclusion of HIV-negative participants in the study sample, and not MSM population in the study sample.

-Study characteristics: “Studies’ characteristics” would be a more appropriate title for this section as it describes the characteristics of the studies included in this review, and not the characteristics of the review itself.

Dear reviewer #1, The title “Studies’ characteristics” has been used.

-As previously mentioned, I recommend using the term “MSM living with HIV” or “MSM with HIV”, instead of HIV positive or seropositive MSM (or simply HIV MSM, as in the Discussion, please avoid using this term). I recommend consistency in the use of terms.

Dear reviewer #1, the term “MSM living with HIV” has been used throughout the manuscript, for consistency.

-I also recommend using the term “social” instead of “sociological”, as it is not related to sociology but to society.

Dear reviewer #1, the term “social” instead of “sociological” has been used throughout the manuscript.

-Within the biological factors, it is stated that “ART initiation improved depression”. This means that depression was reduced, it decreased, it was associated with a reduction of depressive symptoms? Perhaps the sentence could be expressed in a clearer way. The same for viral load, it is not clear if it is associated with increments or reduction of depressive symptoms. I suggest expressing the relations between factors and depression in a clearer way, indicating if they are associated with increased or reduced odds of depression. This same recommendation applies for psychological and social factors.

Dear reviewer #1, 

With regards to biological, psychological, and social factors, the relations between factors and depression has been described in a clearer way, indicating if they are associated with increased or reduced odds of depression/ increased or decreased depression/depressive symptoms.

-Within the psychological factors, it is stated that stigma is associated with increased odds of depression. However, it is not mentioned what kind of stigma: HIV-related stigma, stigma related to sexual orientation (being gay or bisexual) or other kind? I recommend clarifying this.

Dear reviewer #1, 

The type of stigma has been described as enacted HIV-related stigma.

-Regarding the CES-D, please revise the correct name of the instrument for greater accuracy.

-Some of the instruments listed measure “presence of depressive symptoms” and not “depression” (which is a specific disorder and diagnosis). That is, they screen for depressive symptoms but do not allow to arrive to a diagnosis of depression. This is the case, for example, of the CES-D. In that sense, it can be questioned that the present study is actually systematizing and reviewing factors associated with presence of depressive symptoms, and not factors associated with depression (which is not the same).

Dear reviewer #1, the brief justification of accepting tools for masurement of depression has been described. For example with regards of CES-D: Following the test objectives, the CES-D provides cut-off scores; for example, a score of 16 or higher can aid in identifying persons who are at risk for clinical depression, with good sensitivity and specificity, as well as a high level of internal consistency. My references as below.

https://www.apa.org/pi/about/publications/caregivers/practice-settings/assessment/tools/depression-scale & https://journals.plos.org/plosone/article?id=10.1371/journal.pone.0155431). 

However, I understood that it is debatable and I will humbly accept your expert decision. 

Discussion

-Among the biological factors, older age was a protective factor. Given the vulnerability that younger people exhibit and that it is also a key population regarding HIV and mental health, this is a result that is worth mentioning and explaining in the discussion section.

Dear reviewer #1.Thank you. The age factor has been discussed.

-In the “enacted stigma” subsection, it is not clear what “same-sex sexual identity” means, I understand that the authors meant simply “sexual identity”. In this section, it should be clarified what kind of stigma studies refer to.

Dear reviewer #1, the term “sexual identity was used” and stigma refers to enacted HIV-related stigma.

-In the “current smoker” section, it is stated that “The number of life-years lost is…, respectively”. However, it is not clear “respectively” to what or who is referring.

Dear reviewer #1, im sorry for the mistake. The correct sentences is “Smoking was linked to more than 60% of fatalities in PLHIV, whereby they lose more life-years due to smoking, with 12.3 years lost to smoking than 5.1 years lost to HIV”.

-There are other social factors that are worth mentioning and discussing which have not been included in the Discussion section, such as unemployment and migration. In general, I recommend expanding the Discussion with the inclusion of other relevant factors that have been identified as determinants of depressive symptoms (e.g., unemployment, migration, age, self-efficacy, social support). Although evidence may be weak or moderate for some of them, they are relevant to be considered and analyzed for theoretical or practical reasons.

Dear reviewer #1, unemployment, born overseas, age, self-efficacy, social support has been added in discussion. 

-Limitations: I would not say “incomparability”, but “limited comparability”, as studies were actually compared.

Dear reviewer #1, “limited comparability” has been chosen.

-The fact that studies from different countries were included not only implies demographic differences between patients. It also implies different cultural and legal contexts, with deep variations in the acceptance of gay and bisexual (GB) identities and rights. This influences the levels of social stigma (one of the main factors identified in this study). Some of these countries have laws protecting GB people’s rights whereas others do not. This should be acknowledged.

Dear reviewer #1, the concern has been acknowledge briefly and concisely.

-“Foreign language” refers to non-English language? Foreign language depends on the native language of the authors and readers. I agree that this choice introduced bias as most studies come from English speaking countries and there is little evidence from other regions. I would explicitly mention this.

Dear reviewer #1, yes I mean non-English language.

-Conclusion: Regarding the first sentence, the determinants that were identified were more than those that are mentioned. I would mention that these were the determinants with the strongest evidence.

Dear reviewer #1, this sentence has been added: “The determinants of depression with the strongest evidence among MSM living with HIV were…”

-Recommendations: I suggest mentioning more explicitly, actions to fight stigma in general (e.g., enactment of laws to protect GB people’s rights) and in healthcare services (e.g., services that acknowledge, affirm, and validate diverse sexual identities). Elimination of stigma is mentioned too broadly in the recommendations perhaps, and it deserves a more detailed and expanded mention as one of the key determinants of depression among MSM.

Dear reviewer #1, your suggestion has been added in recommendation.

-Table 1, study 4: the date is incomplete.

Dear reviewer #1, the date has been inserted.

-Table 1, study 1: the aim is to study predictors of anxiety and generalized stress. What was the role of depression in this study and why was it included if depression was not the outcome?

Dear reviewer #1, Im sorry for the mistake. the objectives was “to identify and compare risk and protective factors for depression, anxiety, and generalized stress” (Heywood & Lyon, 2016).

-In table 2, I recommend placing the studies in the same order as Table 1 to facilitate integration of the information from both tables by the reader, as both tables are actually related.

Dear reviewer #1. The table has been edited for greater sysyemization. The studies in table 2 has been placed in the same order as table 1. 

-According to Table 2, many factors or determinants of depression that were identified in the present review were drawn from the study of Irving et al. (2018). For that reason, one would argue that the integration of results across studies was limited, and that this manuscript is, in some sense, a summary or reproduction of results of other few studies. Relevant factors that are mentioned in other studies (e.g., internalized stigma, which is mentioned in 3 studies) are barely taken into consideration and analyzed in this review (despite being a very important, repeatedly mentioned factor). A greater systematization of findings is required so that this manuscript is not a mere reproduction of results of a few studies, because that adds little value and significant contribution to the field.

Dear reviewer #1. The relevant factors that are mentioned in other studies (e.g., age, unemployment, internalized stigma, born overseas, self-efficacy, social support) has been added.

6. PLOS authors have the option to publish the peer review history of their article (what does this mean?). If published, this will include your full peer review and any attached files.

Dear reviewer #1. Yes, I agree, if published, this will include my full peer review and any attached files.

Do you want your identity to be public for this peer review? For information about this choice, including consent withdrawal, please see our Privacy Policy.

Dear reviewer #1. Yes, I want our identity to be public for this peer review.

Your consideration of accepting this manuscript for publication and cooperation on this matter is greatly appreciated.

Thank you.

Yours sincerely,

Zul Aizat Mohamad Fisal, MD

Faculty of Medicine and Health Sciences, 

Universiti Putra Malaysia

---

## [Editor Report · Decision Letter 1]

15 Feb 2022

Biopsychosocial approach to understanding determinants of depression among men who have sex with men living with HIV: A systematic review

PONE-D-21-24036R1

Dear Dr. Mohamad Fisal,

We’re pleased to inform you that your manuscript has been judged scientifically suitable for publication and will be formally accepted for publication once it meets all outstanding technical requirements.

Kind regards,

Omar Sued, MD, PhD

Academic Editor

PLOS ONE
---

## [Editor Report · Acceptance letter]

4 Mar 2022

PONE-D-21-24036R1 

Biopsychosocial approach to understanding determinants of depression among men who have sex with men living with HIV: A systematic review 

Dear Dr. Mohamad Fisal:

I'm pleased to inform you that your manuscript has been deemed suitable for publication in PLOS ONE. Congratulations! Your manuscript is now with our production department. 

Kind regards, 

on behalf of

Dr. Omar Sued 

Academic Editor

PLOS ONE